# The Efficacy of Transvaginal Ultrasound-Guided BoNT-A External Sphincter Injection in Female Patients with Underactive Bladder

**DOI:** 10.3390/toxins15030199

**Published:** 2023-03-04

**Authors:** Cheng-Yen Tsai, Yao-Hung Yeh, Li-Hsien Tsai, Eric Chieh-Lung Chou

**Affiliations:** 1Department of Urology, China Medical University Hospital, Taichung 40447, Taiwan; 2School of Medicine, China Medical University, Taichung 40402, Taiwan; 3Department of Radiology, Kaohsiung Veterans General Hospital, Kaohsiung 81362, Taiwan; 4School of Medicine, Chung Shan Medical University, Taichung 40201, Taiwan

**Keywords:** botulinum toxin A, external sphincter injection, transvaginal ultrasound guidance, underactive bladder, precision medicine

## Abstract

Owing to the diverse treatment outcomes after a botulinum toxin A (BoNT-A) injection to the external sphincter, this study aimed to develop a new technique: an ultrasound-guided BoNT-A external sphincter injection. This single-center prospective cohort study was conducted at a tertiary medical center in Taichung, Taiwan. From December 2020 to September 2022, 12 women were enrolled. The patients were evaluated for lower urinary tract syndrome using patient perception of bladder condition (PPBC), international prostate symptom score (IPSS), uroflowmetry, post-void residual volume (PVR), cystometry, and external sphincter electromyography. We evaluated the patients the day before surgery and 1 week after the BoNT-A injection. For the patients requiring self-catheterization, we recorded the number of times they required clean intermittent catheterization (CIC) per day before the procedure and 1 month after the procedure. The IPSS, PPBC, and PVR were significantly better after the transvaginal ultrasound-guided BoNT-A external sphincter injection. The number of times the patients required daily CIC was also reduced after the injection. Only one patient developed de novo urge urinary incontinence. Our results demonstrated that a transvaginal ultrasound-guided BoNT-A injection was efficacious and safe in the treatment of underactive bladder.

## 1. Introduction

A botulinum toxin A (BoNT-A) injection is an increasingly common treatment modality for patients with voiding dysfunction. However, patient outcomes after treatment and their satisfaction with treatment vary widely. Most previous studies have focused on the physiology and pathology of the urinary bladder after the injection, such as idiopathic detrusor overactivity and overactive bladder syndrome. For decades, botulinum toxin has been shown to be effective against both conditions and to significantly improve the quality of life of patients [1]. 

Relatively few studies have investigated the use of BoNT-A injection therapy to the external sphincter, and these studies sometimes reported inconsistent results. Kao et al. summarized the outcomes of subjective parameters for patients diagnosed with detrusor sphincter dyssynergia (DSD) after a sphincter BoNT-A injection, and improvements of up to 61–88% were reported in clinical manifestation. They also found that a 100-unit BoNT-A injection to the external sphincter resulted in a 79–96% improvement in the clinical or urodynamic parameters in patients with a poor relaxation of the external urethral sphincter, with a patient satisfaction rate of 47–52% [2]. In patients with Fowler syndrome, Fowler et al. reported no improvements in the voiding function after a 200-unit BoNT-A injection to the external sphincter [3]. However, Panicker et al. reported an improvement in the objective parameters of Qmax, the post-void residual volume (PVR), and urethral pressure profile at 10 weeks after a BoNT-A urethral sphincter injection in patients with Fowler syndrome [4]. To the best of our knowledge, most current treatments are direct injections of BoNT-A into the external sphincter of the urethra. So far, there is no definitely therapeutic guideline indicating the proper injection site and the dosage. Whether this method can deliver the drug to the urethral sphincter is also still unknown. We hypothesize that the difficulty to obtain consistent outcomes between these previous studies was due to the lack of a precise injection method.

Underactive bladder (UAB) is defined by the International Continence Society as a symptom complex characterized by a slow urinary stream, hesitancy, and straining to void, with or without a feeling of incomplete bladder emptying, sometimes with storage symptoms [5,6]. The diagnosis is based on clinical manifestations, whereas the diagnosis of the detrusor underactivity (DU) is strictly according to urodynamic studies [7]. DU is “a contraction of reduced strength and/or duration, resulting in prolonged bladder emptying and/or a failure to achieve complete bladder emptying within a normal time span”, according to the International Continence Society definition [5,6]. DU frequently occurs with UAB in many patients [5,6,7,8,9]. 

The efficacy of a BoNT-A injection to the external sphincter has been proven in patients with UAB and DU in previous studies. Lee et al. reported a successful outcome rate of up to 59.1% after a urethral BoNT-A injection in male patients with DU based on clinical diagnosis and a videourodynamic classification [10]. In a study of 60 patients with DU by Jiang et al., the patients received a total urethral sphincter injection of 100 units of BoNT-A, and 60% of the patients had good outcomes. However, 12 patients developed de novo urinary incontinence after treatment (four patients reported stress urinary incontinence and eight patients reported urgency urinary incontinence) [11]. With regard to UAB, Chen et al. found that a BoNT-A external sphincter injection could decrease the amount of post-void urine. Of the 20 patients who needed daily clean intermittent catheterization (CIC) before the injection, 75% could self-void 1 month after treatment, and 25% could void without CIC [12]. 

The female external sphincter is very thin. Macura et al. measured the thickness of the external sphincter from the magnetic resonance images of 23 continent volunteers and reported a thickness of 2.06 ± 0.41 mm with surrounding connective tissue [13,14]. Thus, we believe that it is difficult to inject blindly into the striated muscle of the external sphincter.

We hypothesized that if BoNT-A could be accurately injected into the urethral sphincter, the treatment effect in improving lower urinary tract symptoms and patient satisfaction would be elevated. Therefore, we propose a new treatment method to accurately identify the external urethral sphincter using real-time ultrasound guidance, thereby allowing for the precise injection of BoNT-A into the external urethral sphincter. In our study, we hope to discover the optima injection dose and the injection site of the external sphincter.

## 2. Results

From December 2020 to September 2022, a total of 12 women with a mean age of 56.5 ± 16.7 (range, 18–88) years were enrolled. Details of the patient characteristics, including their age, body mass index, length of UAB history, and underlying disease before treatment, are shown in Table 1. All patients in the group experienced hesitancy and a weak urinary stream, which led to the need for abdominal straining during urination. Ten patients reported a residual urine sensation after voiding, and eight experienced nocturia more than three times per night. Uroflowmetry tests conducted before the BoNT-A injection showed a decrease in both the maximum and average flow rates, prolonged voiding time, and an intermittent interrupted pattern in six patients. Cystometry and external sphincter electromyography revealed a decrease in the detrusor contractility. The average post-void residual volume for the patient group prior to treatment was 279.92 mL, and the average voided volume before treatment was 147.16 mL (Table 2).

Seven patients had underactive bladder syndrome with incomplete bladder emptying and relied on CIC before the BoNT-A external sphincter injection. We averaged the post-void residual volume of patients who performed self-catheterization after attempting self-voiding one day before the treatment. The average post-void residual volume before the injection of this group was measured up to 394 mL (Table 3). 

The international prostate symptom score (IPSS) was 21.83 before treatment and 13.33 after treatment, and the difference was statistically significant (*p* = 0.002, Table 2). For our patients, the most obvious improvements were weak streaming, frequency, and nocturia. The PVR was 279.92 mL before treatment and 76.0 mL after treatment, and the difference was also statistically significant (*p* = 0.016, Table 2). The voiding volume showed a significant improvement from a 146.16 mL to 239.50 volume. When we focused on the subjective feelings of the patients, the patient perception of the bladder condition score (PPBC) and quality of life on the IPSS before and after 1 week of management were also both significantly different (Table 2). We performed a urodynamic test including uroflowmetry one week after the injection. The voiding volume and the velocity of both the maximum flow rate and average flow rate improved. However, we did not routinely apply cystometry and external sphincter electromyography to every patient due to the invasive property. The patient often refused to do the test again during the outpatient visit. No urinary tract infections (UTIs) or persistent gross hematuria were noted for more than 2 days, and there were no cases of urethral hematoma. One patient (8.3%) complained of severe urge urinary incontinence that required the use of pads after the external sphincter injection. 

In the seven patients who needed daily CIC, the PVR was 394 mL before treatment and 107.85 mL after treatment, and the difference was significant (*p* = 0.005, Table 3). The number of times the patients required daily CIC before and after 1 month of treatment also significantly reduced from 4 to 1.42 (*p* = 0.019, Table 3). Two patients could void by themself smoothly without lots of post-void residual volume after receiving the BoNT-A injection, and they did not require daily CIC anymore.

## 3. Discussion

In this study, we found that an ultrasound-guided external sphincter injection significantly improved lower urinary tract symptoms in patients with UAB. Both the IPSS and PVR showed promising improvements. The subjective assessment scales, including the PPBC and quality of life on the IPSS, also showed significant improvements one week after the treatment. The main subjective outcome reported by the patients was an improvement in their ability to void. Prior to treatment, they experienced difficulty emptying their bladder due to bladder underactivity, which required significant abdominal effort and resulted in limited urine output. Some patients even required catheterization. Following treatment, they were able to void with less effort and greater ease, which was reflected in the objective measures of an increased voiding volume and decreased post-void residual urine volume. Moreover, the number of times the patients required daily CIC in those who needed it significantly decreased along with the amount of PVR. One patient (8.3%) developed de novo urge urinary incontinence, and this complication has also been reported in previous studies [11]. Kuo et al. reported inconsistencies between patient satisfaction and urodynamic findings due to the increased risk of incontinence after a BoNT-A external sphincter injection in patients with DSD [15,16]. In our study, this patient’s PPBC score and QoL on the IPSS increased from 2 points to 3 points, while the IPSS increased from 17 points to 20 points. She suffered from urgency sensation and found it difficult to postpone urination after the BoNT-A injection.

The female external urethral sphincter is located in the distal two-thirds of the urethra and is composed of striated muscle. At the proximal portion (the mid-urethra), a horseshoe is formed around the female urethra (Figure 1.), where the closing pressure is highest [17,18,19,20]. The striated urethral sphincter makes up 20% to 60% of the total length of the urethra. 

The female external sphincter is very thin, with a thickness of only about 2 mm with surrounding connective tissue [13,14]. Therefore, if a blind injection method is used, it is difficult to accurately achieve the correct position. In this study, we performed pelvis MRI to our patient to identify the structure of the female urethra more clearly. In this 67-year-old patient, the MRI showed a thin external sphincter with a thick connective tissue nearby in the mid urethra (Figure 2). Theoretically, the connective tissue adjacent to the sphincter can be reached using ultrasound positioning. Despite the fact that the goal is to be able to accurately inject into the external sphincter, as this layer of muscle is very thin, it is not easy to inject BoNT-A even with ultrasound guidance. Some of the medication may be in adjacent connective tissue, meaning it may reach the external sphincter through a diffusion mechanism of BoNT-A [21,22]. Previous studies have well documented that the striated urethral sphincter is thin and can be located at a distance outside the urethral lumen. However, clinical outcomes with the injection method have been inconsistent according to previous research. It is uncertain whether the injected botulinum toxin reaches the external sphincter as intended. Our study, which used transvaginal ultrasound, found that without proper guidance, the injected medication may deviate from the intended target. Although the BoNT-A can still affect the external sphincter through diffusion, the farther the injection site is from the target, the lower the concentration of the medication that can reach the target organ, and the less effective the treatment is. Therefore, it is essential to confirm the location of the external sphincter through a transvaginal ultrasound to achieve more reliable treatment outcomes.

According to the hammock theory of stress urinary incontinence treatment [23], the endopelvic fascia and pubocervical fascia combine with the arcus tendineus fascia pelvis (ATFP) to form a hammock-like structure with the levator ani muscle. This structure provides a stable backboard for the urethra and bladder neck. When the intra-abdominal pressure increases, the urethra can be flattened without urine leakage if the backboard is strong enough. Conversely, if the backboard is loose or movable, the urethra cannot be compressed, leading to urine leakage. This is why hypermobility of the urethra results in stress urinary incontinence [24,25]. Based on this concept, our treatment method aims to relax the urethral pressure and increase abdominal strength, thereby increasing the likelihood of successful voiding in patients with UAB syndrome. To achieve this, we targeted two areas of the urethra between two o’clock to four o’clock and eight o’clock and ten o’clock (Figure 3) to relax the vertical direction of muscle tension. Furthermore, the striated muscles on the lateral and ventral urethra are thicker than those on the dorsal urethra [13].

### Regenerate Response

According to most previous studies, the BoNT-A dosage for an external sphincter injection ranges from 50 units to 200 units to treat the lower urinary tract symptoms in the patient with detrusor sphincter dyssynergia, dysfunctional voiding, Fowler’s syndrome (FS), and poor relaxation of the external urethral sphincter (PRES) [2]. Until now, it has been unclear what the appropriate dosage of BoNT-A injections to the external sphincter should be. In previous treatments, most studies used 100 units. However, Kao et al. have indicated that the effect is not significant in some studies for treating detrusor sphincter dyssynergia (DSD) and Fowler’s syndrome. In our study, the objective is to relax the external sphincter, which enables patients to urinate with abdominal pressure. It is hoped that a noticeable effect of relaxing the external sphincter can be achieved. Therefore, before conducting this study, we carried out a pilot study and found that the effect of using 200 units was more significant. Consequently, this dosage was adopted in our study. Regarding a cosmetic BoNT-A injection, Yi et al. conducted a review on the use of BoNT-A injections in the platysma muscle to treat platysmal band and lower facial lifting. The authors concluded that a total of 40 units of BoNT-A should be injected at 20 points on both sides of the platysma band. For a jawline injection, they suggested a total of 40 units for 20 points with a subdermal injection [26]. Doan et al. published a systematic review and meta-analysis to evaluate the efficacy and optimal dose of BoNT-A in the post-stroke lower extremity spasticity (PLES). They found that doses of approximately 300 U of BoNT-A could be the preferable dosage for spastic plantar flexors, which is the most common pattern of PLES [27]. Interestingly, in our study, we observed that after injecting large doses of BoNT-A into a relatively small muscle group, the women’s external sphincter, the patients showed a significant improvement in lower urinary tract symptoms accompanied by a lower incidence of side effects such as incontinence. Therefore, the interaction of the external sphincter with a BoNT-A injection is worthy of further consideration.

Juan et al. have demonstrated that there is variability in muscle weakness following a BoNT-A injection with the same concentration. They propose that the variation in sensitivity to chemodenervation may be related to differences in muscle-specific fibers, particularly the fast-twitch and slow-twitch muscle fibers. It is noteworthy that most of the external sphincter is comprised of slow-twitch muscle fibers. Mechanical properties in rabbit knee extensors have demonstrated that the degree of muscle paralysis following a BoNT-A injection is not uniform and may depend on a variety of physiological and biophysical factors, including the force frequency and force length of the muscle [21]. Furthermore, the complex mechanism of continence, which involves the support of pelvic flow, the importance of the pubourethral ligament, and the closure of the urethra, suggests that other structural factors are involved in preventing urine leakage. Our treatment targets patients with underactive bladder rather than detrusor sphincter dyssynergia. The difficulty in voiding experienced by this group is due to inadequate detrusor muscle contraction, rather than inappropriate involuntary urethral sphincter contraction. This may also explain why most of our patients did not develop severe urgency or stress incontinence.

Based on previous studies, we used a dilution ratio of 4 mL per 100 units of BoNT-A [28]. During our operation, we diluted 200 units of BoNT-A to 8 mL using normal saline. However, we were concerned that injecting 8 mL of liquid into a small region could cause local edema and increase the risk of urine retention. To address this concern, we decided to keep the urethral Foley catheter in place for one day after the operation. We removed the catheter the next morning after the BoNT-A external sphincter injection, and there were no episodes of urine retention in our study. The optimal dilution ratio for BoNT-A injections remains unclear and requires further investigation. In our study, we injected BoNT-A at 16 points, with 0.5 mL at each point, along the 2–4 o’clock and 8–10 o’clock directions of the external sphincter. Juan et al. found that the size and region of the paralyzing field are significantly influenced by multiple-point injections along the target muscle, rather than a single bolus injection. After a single bolus injection of BoNT-A, its effects may spread beyond the boundaries of the target muscle. Conversely, if the total dose is distributed in smaller quantities and injected along the muscle, the biological effect can be confined to the target muscle [21]. A BoNT-A injection into the external sphincter has mainly been used for patients with DSD [15,16,29]. However, the number of patients with UAB syndrome is expected to increase because of our aging society. Yu et al. reported a prevalence rate of DU of 25–48% in elderly men and 12–24% in elderly women with lower urinary tract symptoms [30]. Many middle-aged and older women have difficulty voiding because of UAB. The current therapeutic strategies include CIC, drugs (cholinesterase inhibitors, alpha-blockers, muscarinic agonists), and surgery (transurethral resection of the prostate, diverticulectomy, cystoplasty, and sacral neuromodulation). However, satisfaction with these treatments is still controversial in patients with UAB [11,31,32]. Therefore, if BoNT-A can be used to improve voiding symptoms through a precise injection method, it would be an excellent therapeutic option. In addition, patients with neurogenic bladder or UAB syndrome can void by straining their abdomen, which is known as the Credé maneuver [33]. Men with UAB syndrome have a low chance of success because they have a prostate, which obstructs the bladder outlet. Women with UAB, on the other hand, have a better chance of emptying urine through abdominal straining. However, if resistance of the bladder outlet can be further reduced using a BoNT-A external sphincter injection, the chances of the patient being able to successfully self-empty urine without the need for CIC are higher. Our patients received 200 units of BoNT-A precisely injected under ultrasound guidance to the correct location. This significantly improved voiding symptoms and reduced the residual urine volume, and for patients requiring self-catheterization, it reduced the number of daily CIC.

This study has several limitations, including the small sample size and short follow-up interval. Although patients returned to our outpatient department one month and three months after the operation and reported stable and satisfactory results, only eight patients completed all evaluations, thus the results at one and three months after the BoNT-A injection were not included in the analysis. Performing EMG during the BoNT-A injection would have been ideal for identifying the external sphincter region and monitoring therapeutic effect, but we were unable to do so due to limited hardware facilities and technical difficulties in the operating room. Furthermore, not all patients underwent cystometry and EMG, as many declined the invasive procedure during outpatient visits. Due to the limited sample size, we have not been able to statistically identify potential reasons for treatment failure. Future larger-scale studies may be able to identify patient factors that contribute to treatment failure. Additionally, our study lacked a control group. In future follow-up studies, we plan to compare the prognosis between ultrasound-guided and blind injection methods.

**Figure 1 toxins-15-00199-f001:**
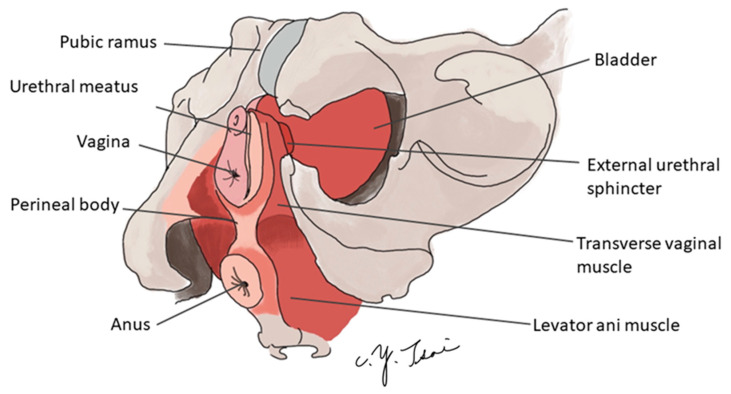
The anatomy of the female urethral external sphincter.

The average length of the female external sphincter is about three centimeters, located in the distal two-thirds of the urethra. In the middle of the urethra, the muscle fiber forms a horseshoe around the female urethra, providing the highest closing pressure. 

**Figure 2 toxins-15-00199-f002:**
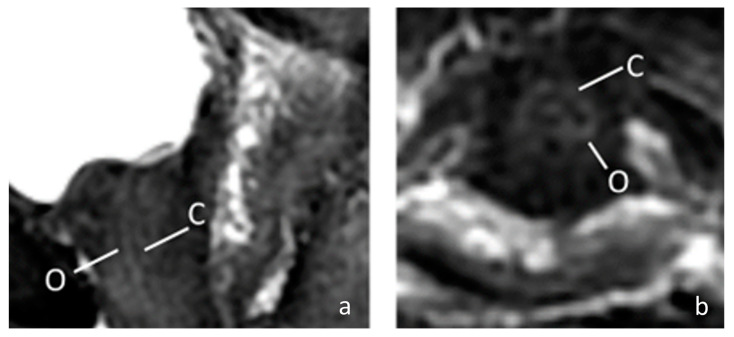
The MRI image of a 67-year-old female urethra. (**a**) Sagittal fat-saturated T2-weighted of female urethra; (**b**) Axial fat-saturated T2-weighted of female urethra. The MRI shows a high signal intensity vascularized connective tissue (C) and low signal intensity outer muscular layer (O). (C = connective tissue; O = outer muscular layer).

**Figure 3 toxins-15-00199-f003:**
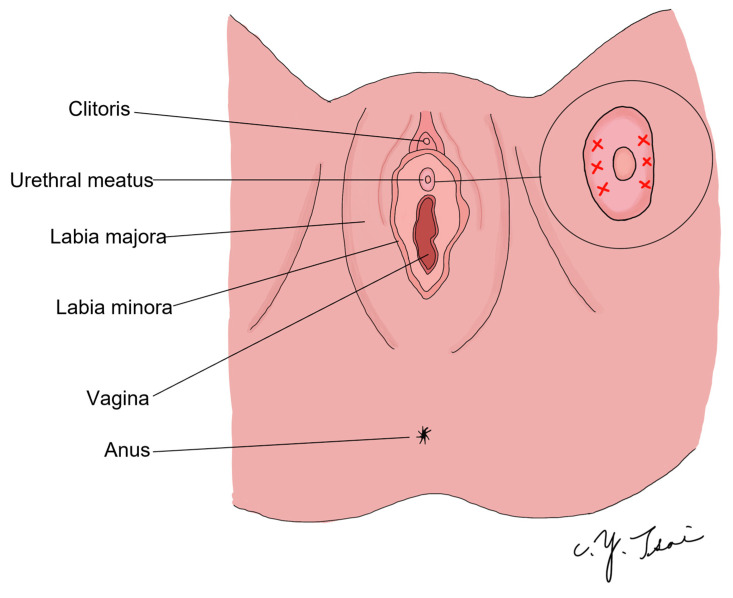
Schematic diagram of the injection site.

We targeted the 2–4 o’clock and 8–10 o’clock directions of the external sphincter (as the red fork area) according to the distribution of the external sphincter and the hammock theory. We injected 200 units BoNT-A at 16 points of the external sphincter in order to relax the vertical direction of muscle tension.

## 4. Conclusions

A transvaginal ultrasound-guided external sphincter injection is a safe and efficacious method to improve lower urinary tract symptoms and reduce the PVR and number of daily CIC in patients with UAB.

## 5. Materials and Methods

This single-center prospective cohort study was conducted at a tertiary medical center in Taichung, Taiwan. We provided the trial protocol and related information to all of the trial participants. All subjects signed informed consent for their inclusion before they participated in the study. The study was approved by the Institutional Review Board of China Medical University Hospital (protocol number MDR-94-IRB-083).

From December 2020 to September 2022, 12 women were enrolled. The patients were evaluated for lower urinary tract syndrome using the patients’ perception of their bladder condition, the IPSS, uroflowmetry, PVR, cystometry, and external sphincter electromyography. We evaluated the patients the day before surgery and 1 week after the BoNT-A injection. For the patients requiring self-catheterization, we recorded the number of times they required CIC per day 1 day before the procedure and 1 month after the procedure. The surgical procedure was performed by a single surgeon (ECL Chou) using a BoNT-A (BOTOX^®^, Allergan, Irvine, CA, USA) ultrasound-guided external sphincter injection.

Patients over 18 years of age who complained of slow urinary stream, hesitancy, and straining to void, with or without a feeling of incomplete bladder emptying, or even relied on CIC, were first screened for UAB syndrome. We included patients with a urodynamic examination showing a staccato-shaped void, a PVR of more than 200 mL, cystometry and external sphincter electromyography showing a reduction in detrusor contractility, and those with no urethral stricture after a cystoscopic examination. The exclusion criteria were vesical stone, recurrent urinary tract infection, malignancy of the urinary tract, and pelvic organ prolapse with bladder outlet obstruction. Patients exposed to radiation therapy or who had previously undergone urethral surgery were also excluded. 

The patient’s age, underlying diseases, weight, and height were recorded. The primary outcome was the change in the PVR or a reduction in the number of CIC for the patients who required self-catheterization. Other outcomes include changes in the PPBC and IPSS.

During the surgery, we performed a cystoscopic examination first. Then, we inserted the 16 French two-way Foley catheter. We infused 150 mL of normal saline into the bladder to make it easier for the ultrasound-guided technique. The BoNT-A 200 units were injected under the real-time method. After the operation, we let the patient keep the urethral Foley catheter and remove it on post-operative day one.

We precisely injected BoNT-A into the external sphincter using the transvaginal ultrasound guidance technique (Figure 4). Under intravenous general anesthesia or spinal anesthesia, the patient was put in the lithotomy position. The external genitalia were prepared and sterilized, and cystoscopy was performed first to check if there was any urethral stricture, vesical stone, bladder tumor, or other lesions. We then inserted one 16 Fr. urethral Foley catheter and infused 150 mL of normal saline to allow for a clearer identification of the external sphincter. Two hundred units of BoNT-A were then injected into the external sphincter under echo guidance (Figure 5). We targeted the 2–4 o’clock and 8–10 o’clock directions of the external sphincter (Figure 3). We diluted 200 units of BoNT-A to 8 cc with normal saline, 0.5 cc per injection to different parts of the external sphincter. After the operation, we let the patient keep the urethral Foley catheter for one day. We evaluated the patient with the PPBC and IPSS 1 day before surgery after hospitalization. We then re-evaluated the patient with the PPBC, IPSS, and PVR 1 week after the operation when the patient visited our outpatient department. For patients who required CIC, we recorded the number of daily CIC and the volume of CIC 1 month after the BoNT-A injection. If the patient could void by herself, the voiding volume would be recorded, too.

We initially located the area of urethral thickening through the transverse view (Figure 5a). To further pinpoint its position in the mid-urethra extending to the distal two-thirds of the urethra, we utilized the sagittal view (Figure 5b). Subsequently, we administered BoNT-A injections into the external sphincter and surrounding connective tissue in the 2–4 o’clock and 8–10 o’clock directions, covering approximately the distal two-thirds of the urethra. (Figure 5c,d).

All comparisons of the patient outcomes were assessed using the paired sample *t*-test. All statistical assessments were performed by two-sided analysis, and significant differences were considered at a *p*-value < 0.05. SPSS version 19.0 (SPSS Inc., Chicago, IL, USA) was used for all statistical analyses.

## Figures and Tables

**Figure 4 toxins-15-00199-f004:**
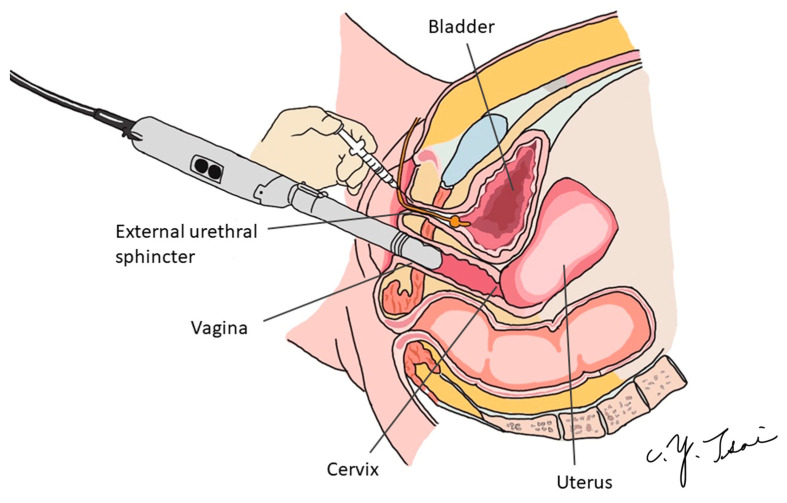
Schematic diagram of the surgery.

**Figure 5 toxins-15-00199-f005:**
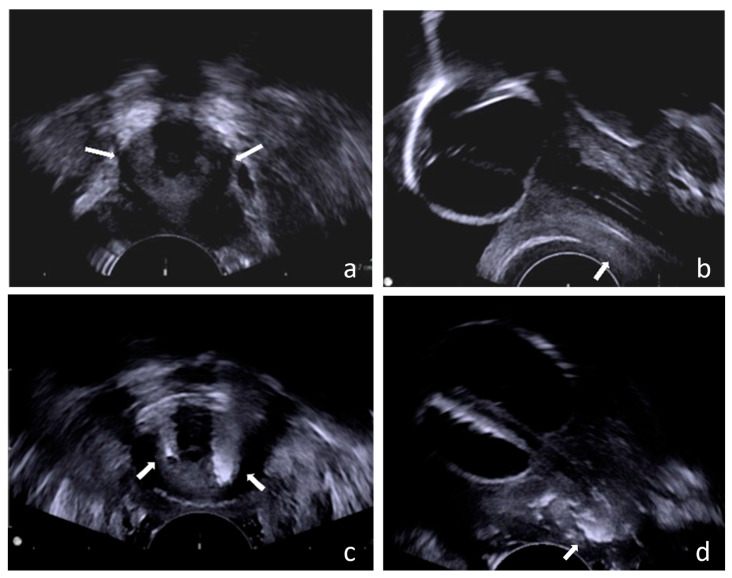
Transvaginal ultrasound guidance technique. We used a transvaginal ultrasound to locate the external urethral sphincter in women. (**a**) Transverse view of the female urethra, the white arrow indicates the location of the external sphincter; (**b**) Sagittal view of the female urethra, the white arrow indicates the location of the external sphincter; (**c**) After the injection of botulinum toxin, the transverse view of the female urethra, the white arrow indicates the location reached by the medication after injection; (**d**) After the injection of botulinum toxin, the sagittal view of the female urethra, the white arrow indicates the location reached by the medication after injection.

**Table 1 toxins-15-00199-t001:** Patient characteristics.

	Patient
Age (median)	56.5 ± 16.7
BMI, median (kg/m^2^)	24.98 ± 3.84
PVR (mL)	279.92 ± 135.10
Length of UAB history, *n* (%)	
<5 years	3 (25%)
5–10 years	2 (16.67%)
10–20 years	5 (41.67%)
>20 years	2 (16.67%)
Underlying disease, *n* (%)	
Recurrent UTI	7 (58.33%)
Type II DM	3 (25%)
SLE	1 (8.33%)
Sicca syndrome	1 (8.33%)

BMI = body mass index; DM = diabetes mellitus; SLE = systemic lupus erythematosus.

**Table 2 toxins-15-00199-t002:** Assessment score and PVR before and after the external sphincter BoNT-A injection.

	Before BoNT-A Injection	After BoNT-A Injection	*p* Value
IPSS	21.83	13.33	0.002
PPBC	3.58	1.75	0.001
QoL on IPSS	3.92	1.83	<0.001
PVR (mL)	279.92	76	0.016
VV (mL)	147.16	239.50	0.003

IPSS = international prostate symptom score; PPBC = patient perception of bladder condition score; QoL on IPSS = quality of life on IPSS; PVR = post-void residual volume; VV = voided volume.

**Table 3 toxins-15-00199-t003:** PVR and CIC times before and after the external sphincter BoNT-A injection.

	Before BoNT-A Injection	After BoNT-A Injection	*p* Value
PVR (mL)	394	107.85	0.005
CIC times/day	4	1.42	0.019

PVR = post-void residual volume; CIC = clean intermittent catheterization.

## Data Availability

The data presented in this study are available on request from the corresponding authors. The data are not publicly available due to the privacy protection policy.

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
