# Peer review of "The Efficacy of Transvaginal Ultrasound-Guided BoNT-A External Sphincter Injection in Female Patients with Underactive Bladder"

_toxins, 2023, doi:10.3390/toxins15030199_

Round 1

Reviewer 1 Report

This paper was report with new precision injection of BOTOX for external sphincter.This is a treatment that has the potential to become gospel for DU patients.Basically, no major problems are seen, but some points need to be clarified.

2. Results

It is clinically important that the number of CICs has decreased, but since this study is preliminary, we believe that data on urinary function should be disclosed as much as possible. For patients who are able to void, it is necessary to describe the volume of voided urine before and after the BOTOX procedure in comparison with the volume of residual urine.

3. Discussion

Since we are discussing DU, it would be better if there was a description of micturition pressure (Pdet). If PFS has not been performed, it should be noted in the discussion.

5. Materials and Methods

"We included patients with a urodynamic examination showing a staccato-shaped void, a PVR of more than 200 ml, cystometry and external sphincter electromyography showing reduce of detrusor contractility, and those with no urethral stricture after a cystoscopic examination."

The author should be describe the results of external sphincter electromyography. It is important to know how many of the enrolled patients have confirmed DSD.

Author Response

Dear reveiwer,

Thank you for your advice and generous encouragement, which motivated our team greatly. The following are the revision of our manuscript according to your comments:

We described the volume of voided urine before and after the BoNT-A procedure in comparison with the volume of residual urine in Table 1, The finding of the urodynamic test including the results of external sphincter electromyography described from the line 97 to line 102.

Reviewer 2 Report

The authors describe a technique of BotulinumtoxinA sphincter injection with ultrasound guidance.  This is a new technique and basically worth publishing.

Some remarks:

Title:

“Precision Medicine: The Efficacy of Transvaginal Ultrasound-Guided BoNT-A External Sphincter Injection in Female patients with Underactive Bladder.” This is a too pompous title, one cannot suggest this technique as more precise if a comparative study is missing so far. “Precision Medicine” must be deleted.

Abstract:

OK

Introduction:

Line 35 : “Kuo et al. summarized the outcomes of subjective parameters for patients diagnosed with detrusor sphincter dyssynergia (DSD)”. This is not the usual way to reference, please use the first author Kao.

Line 37: “… and reported improvements of up to 61%-88% in clinical manifestation [2]. They also found that a 100-unit BoNT-A injection to the external sphincter resulted in a 79%-96% improvement in clinical or urodynamic parameters.”   But in line 48 you say: “We hypothesize that the failure to obtain good and consistent outcomes in these previous studies was due to the lack of a precise injection method”.   But the results were not too bad, weren`t they? You call it failure. And Kuo injected always the sphincter directly, as far as I am informed.

Altogether, the introduction is too long and busy. Shorten this part by at least a third and leaving away figure 1 (which does not contribute additional or new information), would improve the readers alertness.

Results:

Line 93: Leave table 1 away: you mentioned the age in the text, the PVR is doubled in table 2, and you didn`t analyse any influence of the weight of the patients on the outcome.

Line 103: the p-value is different from that in the table (??)

Line 100-103: leave these lines away – any of these information were given in table 2.

Nothing is said about the results in the EMG- measurements and the cystometry, and – very interesting – the uroflowmetry after injection, data supply would be mandatory.

Table 3: Did the patients perform spontaneous voiding or CIC? Or both? If patients with an underactive bladder perform CIC 4 times daily, they usually do not void spontaneous.

Did you measure the PVR all 4 times over the day? Only once?

Furthermore, one inclusion parameter was PVR > 200, but the PVR was only 199.08 (and this is not really an amount to put patients on CIC, isn`t it?).

M&M section:

Why did you do the followup so early? Are there also followup data at a later timepoint?

I would leave figure 4 away. Figure 3 a-d is very illustrative.

Author Response

Dear reviewer,

Thank you for your advice and generous encouragement, which motivated our team greatly. The following are the revision of our manuscript according to your comments:

“Precision Medicine: The Efficacy of Transvaginal Ultrasound-Guided BoNT-A External Sphincter Injection in Female patients with Underactive Bladder.” This is a too pompous title, one cannot suggest this technique as more precise if a comparative study is missing so far. “Precision Medicine” must be deleted.

  • We had deleted the "Precision Medicine" In our title.

Line 35 : “Kuo et al. summarized the outcomes of subjective parameters for patients diagnosed with detrusor sphincter dyssynergia (DSD)”. This is not the usual way to reference, please use the first author Kao.

  • We had corrected the reference method, thanks for your reminder.

Line 37: “… and reported improvements of up to 61%-88% in clinical manifestation [2]. They also found that a 100-unit BoNT-A injection to the external sphincter resulted in a 79%-96% improvement in clinical or urodynamic parameters.”   But in line 48 you say: “We hypothesize that the failure to obtain good and consistent outcomes in these previous studies was due to the lack of a precise injection method”.   But the results were not too bad, weren`t they? You call it failure. And Kuo injected always the sphincter directly, as far as I am informed.

  • We modified the following sentence of “We hypothesize thatthe failure to obtain good and consistent outcomes in these previous studies was due to the lack of a precise injection method” to “We hypothesize that the difficulty to obtain consistent outcomes between these previous studies was due to the lack of a precise injection method.” Because we didn’t mean the failed result of the previous studies.

Altogether, the introduction is too long and busy. Shorten this part by at least a third and leaving away figure 1 (which does not contribute additional or new information), would improve the readers alertness.

  • We move some content and figure 1 to discussion to shorten the introduction. However, we decide to keep Figure 1 and Figure 4 owning to the special issue that we submit is focused on “Clinical and Anatomical Perspectives of Botulinum Neurotoxin”. Thus we favor keeping the content to describe the anatomy related to the BoNT-A injection.

Results:

Line 93: Leave table 1 away: you mentioned the age in the text, the PVR is doubled in table 2, and you didn`t analyse any influence of the weight of the patients on the outcome.

  • Thanks for your advice, We had left table 1 away.

Line 103: the p-value is different from that in the table (??)

  • Thanks for your advice, this is indeed our mistake and has been corrected in the text.

Nothing is said about the results in the EMG- measurements and the cystometry, and – very interesting – the uroflowmetry after injection, data supply would be mandatory.

  • We had added the finding of EMG and cytometry after the injection in line 125. However, we didn't routinely apply cystometry and external sphincter electromyography to every patient due to the invasive procedure. The patient often refused to do the test again during the outpatient visit. We had discussed it in the limitation part.

Table 3: Did the patients perform spontaneous voiding or CIC? Or both? If patients with an underactive bladder perform CIC 4 times daily, they usually do not void spontaneously.

  • There is seven patient who needs daily CIC more than one time.

Did you measure the PVR all 4 times over the day? Only once?

  • We educate the patient to record the PVR before and after the injection. We document the average PVR of the day in our study.

Furthermore, one inclusion parameter was PVR > 200, but the PVR was only 199.08 (and this is not really an amount to put patients on CIC, isn`t it?).

  • Thanks for your advice, this is indeed our mistake and has been corrected in the text.

M&M section:

Why did you do the followup so early? Are there also followup data at a later timepoint?

  • The patient returned to our outpatient department one month and three months after the operation and obtained stable and satisfactory results. However, due to incomplete data, only eight patients returned to the hospital regularly and completed all the evaluations, so the result of one month and three months after the BoNT-A injection were not included in the analysis of our study. We had discussed this point in the limitation part.

I would leave figure 4 away. Figure 3 a-d is very illustrative.

  • Thanks for your advice, we favor keeping the content and figure to describe the anatomy related to the BoNT-A injection.

We appreciate to your comments.

Round 2

Reviewer 2 Report

No further comments

Author Response

Thank you sincerely for your guidance, which has helped us to improve our paper.